# Inorganic Nanomaterial for Biomedical Imaging of Brain Diseases

**DOI:** 10.3390/molecules26237340

**Published:** 2021-12-03

**Authors:** Wenxian Du, Lingling Zhou, Qiang Zhang, Xin Liu, Xiaoer Wei, Yuehua Li

**Affiliations:** 1Department of Radiology, Shanghai Jiao Tong University Affiliated Sixth People’s Hospital, Shanghai 200233, China; wx0910@mail.ustc.edu.cn (W.D.); llzhou_joyan@163.com (L.Z.); zhangqiang@shanghaitech.edu.cn (Q.Z.); 2State Key Laboratory of Bioelectronics, Jiangsu Key Laboratory for Biomaterials and Devices, School of Biological Science & Medical Engineering, Southeast University, Nanjing 210096, China; liuxin4552@seu.edu.cn

**Keywords:** biomedical imaging, inorganic nanomaterial, brain disease

## Abstract

In the past few decades, brain diseases have taken a heavy toll on human health and social systems. Magnetic resonance imaging (MRI), photoacoustic imaging (PA), computed tomography (CT), and other imaging modes play important roles in disease prevention and treatment. However, the disadvantages of traditional imaging mode, such as long imaging time and large noise, limit the effective diagnosis of diseases, and reduce the precision treatment of diseases. The ever-growing applications of inorganic nanomaterials in biomedicine provide an exciting way to develop novel imaging systems. Moreover, these nanomaterials with special physicochemical characteristics can be modified by surface modification or combined with functional materials to improve targeting in different diseases of the brain to achieve accurate imaging of disease regions. This article reviews the potential applications of different types of inorganic nanomaterials in vivo imaging and in vitro detection of different brain disease models in recent years. In addition, the future trends, opportunities, and disadvantages of inorganic nanomaterials in the application of brain diseases are also discussed. Additionally, recommendations for improving the sensitivity and accuracy of inorganic nanomaterials in screening/diagnosis of brain diseases.

## 1. Introduction

The rising incidence of brain diseases over the past few years has become the most common cause of disability and death worldwide, placing a heavy burden on families and societies. Brain diseases can be divided into many types. The first category is tumors of the brain, including primary tumors of the brain, such as gliomas. It also includes brain metastases from secondary brain tumors, such as lung, breast, prostate, and colorectal cancer [1]. The second category is brain trauma, such as concussion, brain contusion and laceration, diffuse axonal injury, various types of intracerebral hemorrhage, and intracranial hemorrhage. The third category is the brain vascular diseases, including ischemic cerebrovascular disease and cerebral aneurysm, hypertensive intracerebral hemorrhage, stroke, and other diseases. In addition, the neurodegenerative diseases caused by the loss of neurons and their myelin sheath, which deteriorate over time and result in dysfunction, are also problematic. For instance, Alzheimer’s disease (AD), Parkinson’s disease (PD), and epilepsy. To some extent, the central nervous system (CNS) or peripheral nervous system (PNS) is irreversibly damaged by these diseases. Therefore, accurate diagnosis and clinical evaluation of these diseases are very important. However, the current diagnosis method is relatively simple, the resolution is not high, and it is difficult to accurately locate the lesion site, which greatly limits the precise treatment of the disease. So this review introduces different brain diseases and corresponding treatment methods. Then use traditional imaging methods to introduce the imaging applications of inorganic nanomaterials in different brain disease models, hoping to inspire more convenient and accurate imaging methods and integration of diagnosis and treatment (Figure 1).

### 1.1. Brain Disease

#### 1.1.1. Alzheimer’s Disease (AD)

According to the World Health Organisation (WHO) reports that more than 36 million people are suffering from AD. The main pathological features of AD are amyloid β deposition in Senile Plaques (SP) and neurofibrillary tangles (NFTs). The early stage of AD is characterized by memory impairment to a certain extent, and other cognitive impairments (Such as memory loss, aphasia, loss of cognitive ability, and visual space disorders) may occur with the progression of AD [2,3]. The pathogenesis of AD is very complex, and it is difficult to achieve an early diagnosis of the disease. No specific drug for AD has been developed so far, despite the involvement of interdisciplinary researchers. Therefore, how to maintain and improve the cognitive function of the elderly and improve their quality of life in old age has become a hot issue of scientists.

Targeting the central nervous system is the primary treatment for AD. However, the protective and selective role of the blood-brain barrier (BBB) in the central nervous system prevents the drug from effectively passing through the BBB, thus compromising its efficacy [4]. Zhang et al. developed a dual-functional nanoparticle drug delivery system based on a PEGylated poly (lactic acid) (PLA) polymer. They used phage to filtrate two targeting peptides called TGN and QSH that specifically target ligands at the BBB and affinity with Aβ_1–42_. These two targeting peptides could be conjugated to the surface of the nanoparticles to enhance and target delivery to amyloid plaque in the brains of AD model mice (Figure 1a) [5]. In another report, Qu’s group designed an oligomer-specific fluorescent probe based on the hydrophobic regions that are exposed on the Aβ oligomer surfaces, opening a new door to developing a “sense and treat” system for AD therapy. The probe and KLVFF peptide (an Aβ-target peptide) was modified on the surfaces of magnetic nanoparticles (MNP@NFP-pep) to enhance the ability to pass through BBB and recognize Aβ oligomers. Importantly, this complex can detect the Aβ oligomer specifically and realize the wireless deep magnetothermally mediated depolymerization of Aβ oligomer in an alternating magnetic field [6]. Combining nanomedicine with neurodegeneration, as shown in Figure 1b, Cai et al. synthesized hollow manganese Prussian white nanocapsules (HMPWCs), which can achieve the purpose of alleviation of cognitive decline and attenuation of Tau-related pathology [7].

Although the current research on AD is being further advanced, it still needs major breakthroughs. Combining the clinical features of AD, finding specific biological markers and multi-modality imaging examination, establishing accurate early diagnosis technology, and realizing the early diagnosis of the disease to prescribe medicine according to the symptoms can greatly improve the patient’s life treatment and reduce the burden on the family.

#### 1.1.2. Parkinson’s Disease (PD)

Parkinson’s disease (PD) is the second most common neurodegeneration after AD [8]. Genes, aging, and environmental factors play an important role in the development of PD. PD is the gradual loss of Pars compacta dopaminergic neurons, leading to a decrease in dopamine levels, which can lead to severe motor dysfunction. It is characterized by hypokinesia and hyperkinesia, such as myotonia, bradykinesia, and resting tremor [9,10,11]. As the disease progresses, the patient’s exercise capacity will gradually be lost. In the late stage, some patients even suffer from muscle atrophy and the inability to move the joints, which brings heavy physical and psychological pressure to the patients [12]. Therefore, finding effective drugs to treat PD is extremely urgent, which requires scientific researchers to continuously research new drugs and develop new therapies.

At present, the traditional treatment of PD is mainly drug therapy and non-drug therapy [13]. However, drug treatments can only slightly control the patient’s initial symptoms because drugs cannot prevent or delay the degenerative changes of dopaminergic neurons [14,15]. Non-drug treatments include rehabilitation training, diet therapy, gene therapy, and surgery. However, these treatments have not achieved satisfactory results. As a result, multidisciplinary researchers have been committed to the development of new drugs for the treatment of PD.

Sun’s group used gold nanoclusters (AuNCs) as anti-PD drugs for the first time. At the same time, AuNCs can cross the BBB, and the elimination half-life of drug metabolism is 9.317 ± 0.681 h. This research opens up a new way for the development of anti-PD drugs and also expands the application prospects of AuNCs in the field of medicine [16]. In addition to overcoming BBB delivery of drugs, researchers have prepared oral nanoparticles that overcome the gastrointestinal barrier and deliver drugs into the brain to treat PD. As shown in Figure 1c, Chen et al. synthesized a six-arm star polymer, then loaded puerarin by anti-solvent precipitation method, and coated the nanoparticles with D-α-tocopherol PEG_1000_ succinate. Pharmacokinetic studies have found that after oral administration of nanoparticles, the drug concentration in the blood and brain is significantly increased, and the biological half-life is also prolonged. The results showed that the drug-loaded nanoparticles improved PD-related motor dysfunction and reversed the content of dopamine and its metabolites [17].

At present, researchers have conducted extensive studies on the molecular mechanism of PD and have also made progress in gene therapy [18]. Clare L. Parish et al. introduced the suicide gene (HSV-TK) into the transplanted cells and induced the expression of the suicide gene at a specific time point through drug induction. In order to achieve the purpose of reducing the growth of non-target cells and enriching dopaminergic neurons in the substantia nigra compact part [19]. In addition, the development of imaging has further promoted the treatment of PD. Evidence from dopaminergic images and cerebral blood flow/metabolism images reveals the differences in cognitive physiology between healthy people and those with PD [20]. We believe that with the in-depth research on the mechanism of PD and the development of visualized multi-modal treatment methods, PD patients will be cured gradually.

#### 1.1.3. Ischemic Stroke (IS)

Cerebrovascular disease, also known as stroke, has become the second leading cause of death and the third leading cause of disability worldwide [21]. Among them, ischemic stroke (IS) is the disease with the highest disability rate in a single disease, and it is also the main type of stroke in clinical practice. In recent years, the incidence of IS is increasing at an annual rate of 8.7%, which has brought huge mental torture and financial burden to patients and their families. Therefore, it is urgent to develop new treatment methods to enhance the cure of stroke [22]. 

Nowadays, IS is mainly treated by intravenous thrombolytic therapy and intravenous administration [23]. Although thrombolytic therapy as the gold standard can effectively achieve vascular recanalization and reduce the mortality of patients, due to the existence of reperfusion injury, patients often experience further deterioration of brain damage and limited blood perfusion after vascular unblocking. The phenomenon of “no-reflow” makes nearly half of the patients become disabled, which seriously affects the prognosis and quality of life [24,25]. From the intravenous administration of small molecule neuroprotective drugs due to the blocking effect of BBB, the intake of cerebral infarction sites is relatively small, and large doses are required, which can easily lead to kidney function damage and induce side effects such as skin rashes [26,27]. These are clinical unsolved problems of treating stroke.

At the molecular level, the core of the onset of IS is neuronal damage caused by excessive ROS [28]. Therefore, the research of free radical scavengers has become a research hotspot in the neuroprotective treatment of IS. When a brain disease occurs, it is often difficult for drugs delivered by intravenous injection to pass through the BBB to reach the location of the lesion, resulting in a sharp decline in the efficacy of the drug. The rapid development of nano-drug carriers provides the possibility for the development of new targeted drugs.

In recent years, research on brain-targeting carrier materials has become more and more in-depth. Liposomes, polymer nanoparticles, and artificial membrane materials have all shown certain brain-targeting capabilities. These improved MSCs have achieved significant results in mice with IS, which is superior to the therapeutic effect of natural stem cells [29]. In addition, inorganic nanomaterials can be designed as drug delivery carriers of different sizes, shapes, and surface charges. Through surface functionalization and group modification, they can specifically target brain cells to enhance their efficiency across the BBB, prompting the rapid concentration of drugs in the brain injury area, thereby improving the therapeutic effect [30]. From Figure 1d, a non-viral and magnetic field-independent gene transfection strategy is reported. They used ferrous magnetic iron oxide nanochains to improve natural mesenchymal stem cells (MSCs) derived from embryos. The nano-chain can be used as a gene delivery carrier, which realizes high-efficiency gene recombination for MSCs without relying on an external magnetic field, simplifies the current magnetotransfection process based on magnetic nanoparticles, and avoids the use of external magnetic fields that may cause adverse effects on stem cells [29]. This provides new research ideas for improving the efficacy of stem cell therapy.

**Figure 1 molecules-26-07340-f001:**
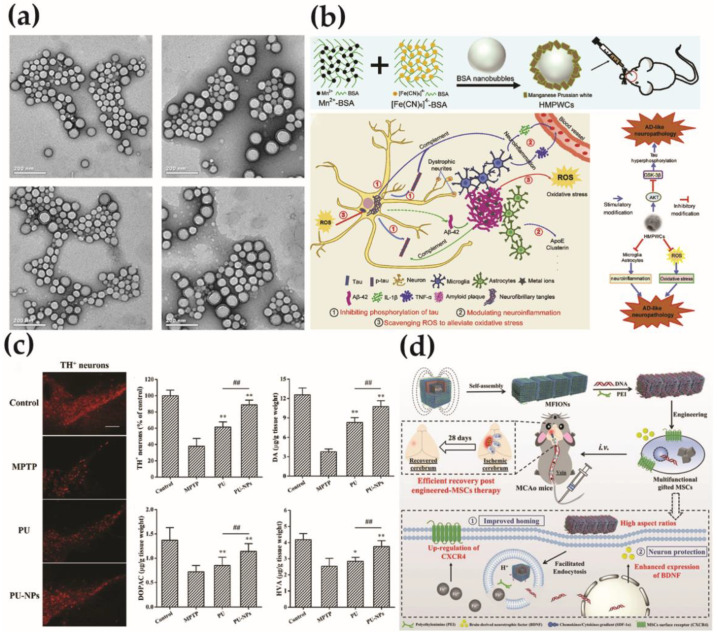
(**a**) Transmission electron micrographs of NP, T_3_-NP, Q_3_-NP, and T_3_Q_3_-NP, respectively, reprinted from [5]. Copyright 2014, with permission from Elsevier; (**b**) schematic illustration of the preparation of HMPWCs and the effect of HMPWCs on the Tau-related AD-like neuropathology, reprinted from [7]. Copyright 2020, with permission from Elsevier; (**c**) PU-NPs reduce neurotoxicity induced by MPTP, reprinted with permission from [17] Copyright (2019) American Chemical Society; (**d**) schematic illustration of MFION-based engineering of MSCs for the recovery post-ischemic stroke, reprinted from [29]. Copyright 2019, with permission from Weily. ** *p* < 0.01 corresponds to different treatments vs. MPTP group; ## *p* < 0.01 corresponds to treatment with PU-NPs vs. with PU group.

#### 1.1.4. Glioma (GBM)

Glioma (GBM) is the most common and aggressive tumor in the central nervous system, and it is still clinically incurable [31,32]. Under the premise of protecting brain function as much as possible, tumor removal to the greatest extent, postoperative concurrent radiotherapy, and chemotherapy is the standard treatment for GBM, but the recurrence rate of patients is still close to 100%. In addition, this treatment has a very poor prognosis, with a median survival time of only 9–12 months and a 2-year survival rate of only 8–12% [33,34,35]. This is largely due to the existence of the BBB, which severely limits the successful delivery of diagnostic and therapeutic molecules (drugs, MR contrast agents, etc.) to the brain, thereby affecting the diagnosis and treatment of GBM [36].

In order to achieve high-efficiency treatment of GBM, Shi et al. designed a small interfering RNA (siRNA) delivery system disguised as red blood cell membrane (RBCm) for gene therapy of GBM. The multifunctional biomimetic siRNA nanocomposite effectively solves the current problems of a short half-life, poor BBB penetration, insufficient tumor accumulation, and poor cellular uptake faced by siRNA, and can realize the efficient treatment of glioma [37]. The combination of Transferrin (Tf) and Tf receptors (TfRs) overexpressed on the surface of BBB endothelial cells is one of the methods to effectively cross the BBB [38,39]. Chen et al. adopted a gentle biomimetic mineralization strategy to grow MnO_2_ nanocrystals in situ on iron-saturated transferrin and further efficiently load the sonosensitizer protoporphyrin. The preparation method well maintains the ability of transferrin itself to cross the BBB and target GBM and endows it with high specific T1-weighted magnetic resonance imaging (MRI) signal enhancement and high specificity in response to the GBM microenvironment. High-efficiency ultrasound dynamic therapy [40]. What is important is that this Tf -based integrated diagnosis and treatment nano-agent has excellent biological safety and versatility, shows good clinical potential, and provides a useful reference for the efficient diagnosis and treatment of a variety of brain diseases.

## 2. Traditional Imaging Techniques

This section may be divided into subheadings. It should provide a concise and precise description of the experimental results, their interpretation, as well as the experimental conclusions that can be drawn.

### 2.1. Magnetic Resonance Imaging (MRI)

MRI, as a commonly used clinical imaging and diagnostic technique, plays an important role in the early diagnosis of diseases. MRI is a physical phenomenon. As an analysis method, it is widely used in physics, chemical biology, and other fields. It was not used for medical clinical testing until 1973. In order to avoid confusion with radiography in nuclear medicine, it is called MRI. The best results are the brain, its spinal cord, heart, and great blood vessels, joint bones, soft tissues, and pelvis. For cardiovascular diseases, it can not only observe the anatomical changes of various chambers, large blood vessels, and valves but also perform ventricular analysis for qualitative and semi-quantitative diagnosis. It can make multiple cross-sectional views with high spatial resolution [41,42].

At present, small molecule gadolinium (Gd) complexes are the main clinical contrast agents. After intravenous injection, the contrast agent is quickly distributed throughout the body through the blood circulation and can spread to the extracellular space, and then is quickly excreted from the body by the kidneys [43]. However, these extracellular contrast agents have problems, such as a short half-life and low relaxation rate, which, in turn, leads to weaker and faster attenuation of the MRI enhancement signal in the body, and it is difficult to maintain high-resolution magnetic resonance imaging. At the same time, unreasonable dose control or excessive use of contrast agents can cause toxic side effects [44]. Therefore, a lot of research has been devoted to the preparation of new contrast agents with high relaxation rates and meeting the basic requirements of contrast agents.

### 2.2. Computed Tomography (CT)

Computed Tomography (CT) uses precisely collimated X-ray beams, gamma rays, ultrasound, etc., together with a highly sensitive detector to scan a certain part of the human body one by one. It has the characteristics of fast scanning time and clear images. It can be used for the inspection of various diseases [45]. In order to obtain more accurate and clear CT imaging effects, CT contrast agents are particularly important. According to the rules of X-ray absorption coefficient, materials with higher density or high atomic number can be used as CT contrast agents [46]. Currently, CT imaging contrast agents used in clinical practice are mainly triiodobenzene derivatives, such as iohexol and iomeprol. However, these small molecule CT contrast agents have problems such as poor targeting and rapid renal clearance during clinical use [47]. Therefore, we must work hard to develop CT imaging contrast agents that are safe and highly targeted as soon as possible.

### 2.3. Fluorescence Imaging (FL)

Fluorescence imaging (FL) is an optical imaging technology that uses fluorescent probes to label cells or specific molecules to generate fluorescent signals under the excitation of external light to achieve imaging [48]. Because of the low cost, convenient operation, quantitative sensitivity, and inherent biological safety of fluorescence imaging, it has been widely used in the field of biological imaging [49,50,51]. Common fluorescent probes mainly include organic fluorescent dyes, fluorescent proteins, and fluorescent lanthanide compounds [52,53,54]. FL has the advantages of high sensitivity and fast imaging. However, in vivo FL faces the problems of biological endogenous fluorescence signal interference, photobleaching, and fluorescence quenching. Therefore, near-infrared fluorescent probes with the advantages of deep imaging penetration depth and good signal-to-noise ratio have become research hotspots [52,55].

### 2.4. Positron Emission Computed Tomography (PET)

Positron emission computed tomography (PET) is an important part of nuclear medicine molecular imaging technology [56]. PET is the only new imaging technology that can display the metabolism of biomolecules, receptors, and nerve mediators in the living body. It has been widely used in the diagnosis and differential diagnosis of various diseases, disease judgment, curative effect evaluation, organ function research and new drug development, etc. It has the advantages of high sensitivity, good specificity, whole-body imaging, and high safety. In recent years, PET has played a more important role in tumors, neurological diseases, and cardiovascular diseases [57,58].

Now, more and more PET tracers can reflect amino acid metabolism, accounting metabolism, and antisense imaging in vivo. However, no PET tracer can target tumors to enter clinical application at home and abroad. Therefore, it is of far-reaching significance to develop a tumor-targeted PET tracer with my country’s independent intellectual property rights to improve the early detection, early diagnosis, and early treatment of tumors in our country.

### 2.5. Photoacoustic Imaging (PA)

Photoacoustic imaging (PA) is a new non-invasive and non-ionizing biomedical imaging method, which surpasses the depth limitation of traditional ballistic optical imaging and the resolution limitation of diffuse optical imaging [59]. The photoacoustic signal generated by the biological tissue carries the light absorption characteristic information of the tissue. By detecting the photoacoustic signal, the light absorption distribution image in the tissue can be reconstructed, which can reflect the morphological structure, pathological characteristics, and metabolism of the biological tissue to a certain extent. PA imaging combines the advantages of high selectivity in pure optical tissue imaging and deep penetration in pure ultrasound tissue imaging to obtain high-resolution and high-contrast tissue images [60,61]. Therefore, it has been more and more widely used in clinical research, especially scientific research [62].

### 2.6. Ultrasound Imaging (US)

Ultrasound imaging (US) is the use of ultrasonic sound beams to scan the human body through the reception and processing of reflected signals to obtain images of internal organs [63]. Ultrasound Doppler is a detection technology that has developed rapidly in recent years. With the advancement of electronics, this method has been widely used in clinical practice. It is of great value in the examination of flow supply, blood supply of space-occupying lesions, and fetal blood circulation [64,65].

As we all know, single-mode imaging methods have their own characteristics, but they also have characteristics, such as low sensitivity and low spatial resolution [66]. Therefore, the gradually developed multi-modal imaging method can more accurately diagnose the location of the lesion and related information. With the development of imaging medicine and multi-modal visualization technology, rapid and accurate diagnosis of major diseases will become possible, laying a good foundation for precise treatment.

### 2.7. Others Imaging Techniques

Magnetic resonance spectroscopy (MRS) is a functional imaging technology that can provide spatially encoded metabolic information and biochemical changes so as to achieve the purpose of non-invasive diagnosis of diseases. At present, MRS is widely used in the diagnosis of central nervous system diseases, especially glioma diseases [67,68]. Besides, magnetic particle imaging (MPI) is a new type of biomedical imaging technology. Different from traditional MRI, MPI realizes direct imaging of the spatial distribution of magnetic particles by directly detecting changes in the responsiveness of magnetic particles in a magnetic field [69].

## 3. The Imaging Applications of Inorganic Nanomaterials in Different Brain Disease Models

In recent years, different nano-formulations have been or are being developed for use in the diagnosis and treatment of brain diseases. Nanomaterials have different shapes, such as spherical, rectangular, and linear morphologies [70]. Nanomaterials can be easily surface-functionalized, which can not only improve the biocompatibility of nanomaterials but also change the charge distribution on the surface of various nanomaterials and achieve a certain degree of targeting or other functional effects [71]. Taking nanomaterials effectively spanning the BBB as an example, the nanomaterials are stable in the blood, have long blood circulation, and cannot cause platelet aggregation. The most important thing is that the modified nanomaterials can achieve the purpose of crossing the BBB at different times through receptor-mediated endocytosis or grafting of some targeting peptides [70,72,73,74].

The increasing number of non-invasive imaging modes illustrates the need for better imaging modes in medical systems. The research of molecular targeted contrast agents has become the key to multimodal imaging. The characteristics of various inorganic nanomaterials-based imaging modalities in terms of spatial resolution and molecular sensitivity are summarized in Table 1. Therefore, the following will use inorganic nanomaterials as the starting point to separately illustrate the imaging applications of non-metal-based nanomaterials and metal-based nanomaterials in different brain disease models.

### 3.1. Nonmetallic-Based Nanomaterials

#### 3.1.1. Carbon-Based Nanomaterials

In recent years, carbon-based nanomaterials have aroused great interest in the fields of neuronal tissue imaging, drug delivery, and electrical sensing [82]. Carbon-based nanomaterials can interact with biological systems at the molecular level and have a high degree of spatial and temporal specificity. They can penetrate the BBB and deliver specific therapeutic drugs, probes, or biological materials to target cells and tissues in the living brain [83].

The elimination and detection of Aβaggregates are essential for the treatment and diagnosis of AD. According to reports, Sun et al. synthesized Nitrogen-doped carbonized polymer dots (CPDs) by hydrothermal method. CPDs can not only effectively inhibit Aβ aggregation and rapidly depolymerize mature Aβ fibers but also realize real-time fluorescence detection of Aβ plaques. As shown in Figure 2a, the green fluorescence signal of ThT-labeled Aβ40 aggregates decreased with the increase in the concentration of CPDs, which was due to the inhibitory effect of CPDs. Additionally, the interaction between CPDs and Aβ40 aggregates resulted in different red fluorescent plaques [84]. In addition to the application of carbon-based nanomaterials in the diagnosis and treatment of AD, they also have great potential in revealing the detailed functions and pathophysiology of the living brain. The combination of nanotechnology and the latest microscopy technology can reveal the complexity of nerve cells and the CNS, which is also conducive to the diagnosis and treatment of brain diseases [85]. For example, amine-functionalized multi-walled carbon nanotubes can exhibit inherent luminescence, and the photon emission falls at 450–650 nm, which is the visible region of the electromagnetic spectrum [86]. Using multiphoton fluorescence technology, intravenously injected multi-walled carbon nanotubes (MWCNTs) can accumulate within 5 min of injection to achieve brain tissue imaging. Although these results show the promise of in vivo brain imaging, the high absorption and scattering of visible fluorescence limits potential deep imaging applications [76]. Therefore, multi-modal imaging should be developed to achieve precise diagnosis and treatment.

**Figure 2 molecules-26-07340-f002:**
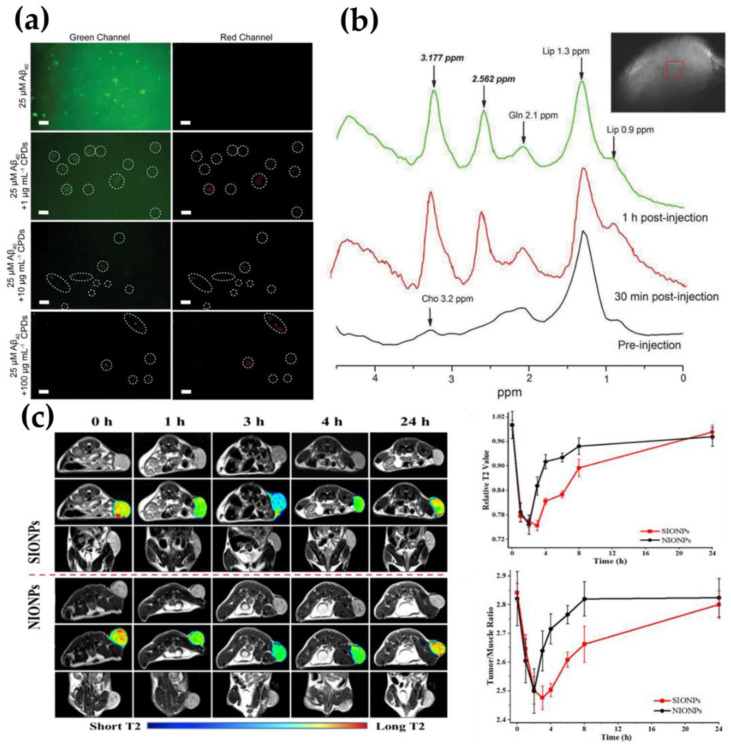
(**a**) FL of Aβ40 incubated with or without CPDs after 56 h at 37 °C, reprinted from [84]. Copyright 2020, with permission from Weily; (**b**) MRS spectra of subcutaneously transplanted glioma before and after the intratumoral injection of 1 mL AMSNs physiological saline solution (10 mg AMSNs/mL, 40 mg AMSNs/kg), reprinted from [77]. Copyright 2018, with permission from Weily; (**c**) in vivo T2-weighted imaging and the corresponding time course of relative T2 signal intensity of tumors, reprinted from [79]. Copyright 2021, with permission from Elsevier.

#### 3.1.2. Silicon-Based Nanomaterials

Silicon is one of the essential trace elements of the human body, accounting for 0.026% of body weight. Silicon is related to the formation of connective tissue and cartilage. Due to the stable structure and good biocompatibility of silicon-based materials, silicon-based nanomaterials as carriers are more and more widely used in the medical field.

Traditional magnetic MRS imaging technology is only limited to the detection of endogenous metabolites and cannot accurately distinguish between GBM and other tumors, such as lymphomas and metastases, which greatly limits its application in disease diagnosis [87,88,89]. Bu’s research group proposed the concept of a contrast agent for functional MRS. By loading exogenous β-alanine on Angiopep-2 (ANG)-modified hollow mesoporous silica spheres (HMSN), the GBM obtained an active targeted and accurate diagnostic function of functional MRS, which found that the GBM lesion area of tumor-bearing mice can show specific β-alanine peaks, while the normal brain tissue areas did not show. Therefore, this new type of contrast agent with a specific enhanced MRS imaging function will be expected to achieve an accurate imaging diagnosis of GBM. As shown in Figure 2b, β-alanine has two characteristic peaks at 2.562 ppm and 3.177 ppm. It is the 2.562 ppm peak that distinguishes it from other substances in the brain and has a special enhanced MRS imaging function, which promotes the rapid development of MRS imaging technology in clinical tumor precise imaging diagnosis [77].

#### 3.1.3. Other Nonmetallic-Based Nanomaterials

In addition to carbon-based silicon-based and other mainstream non-metal-based nanomaterials used in brain disease imaging, some quantum dots and upconversion luminescent nanomaterials also play an important role. This type of nanomaterial has a wider absorption band and a narrower emission spectrum in the visible-near-infrared spectrum. Therefore, it has greater advantages than organic dyes in the imaging of brain gliomas [90,91,92].

### 3.2. Metal-Based Nanomaterials

#### 3.2.1. Iron-Based Nanomaterials

Iron-based nanomaterials have special physical and chemical properties. In particular, Fe_3_O_4_ nanomaterials have multiple crystal forms. Additionally, the morphology of the nanomaterials is also diversified, thus having different properties. So far, iron-based nanomaterials are widely used in electromagnetics, catalysis, biomedicine, environmental protection, and other related fields. MRI is one of the most important clinical diagnostic methods. The currently commonly used contrast agent is Gd-DTPA, but it has problems, such as poor distribution and high prices [93]. In contrast, iron-based nanomaterials have a high specific distribution in living tissues, so they have broad application prospects. In particular, Fe_3_O_4_ and maghemite (γ-Fe_2_O_3_)-based crystalline particles can be transformed into nanomaterials (3–100 nm) with different characteristics by controlling the raw materials, reaction time, and preparation methods. Because both iron oxides contain atomic holes and surface defects, they are easy to be modified for surface functionalization [94,95,96]. Interestingly, when prepared by conventional methods, Fe_3_O_4_ is magnetic, has a cubic inverse spinel structure, and often has tetrahedral and octahedral positions. After reasonable modification, it can be better dispersed to form a magnetic fluid, which can be imaged under the condition of a magnetic field [97].

The distinction between inflammatory mass and malignant GBM has always been a difficult point in clinical diagnosis. Studies have shown that the lateral relaxation time of Fe_3_O_4_ nanoparticles is closely related to their particle size. Fe_3_O_4_ with a large particle size can significantly shorten the T2 lateral relaxation time, thereby increasing the contrast of MRI [78]. The reduction-responsive amphiphilic polymer mPEG-S-S-C_16_ was synthesized by Wu’s group. The encapsulated Fe_3_O_4_ magnetic nanoparticles can enhance its T2 lateral relaxation and identify inflammatory masses and malignant gliomas. It can be seen that the T2 signal is significantly enhanced by the reduction-activated concentrated magnetic resonance contrast agent used in the imaging of GBM in situ (Figure 2c), the T2 time is shortened by about 75.3%, and the T2 signal of the inflammatory mass has no obvious change [79]. 

GBM is the most common brain tumor with a very high fatality rate. It often infiltrates adjacent tissues, and its shape is variable without a definite scope [98,99]. Effective GBM treatment requires sensitive tumor display during surgery and effective postoperative chemotherapy. Unfortunately, the spread and invasiveness of GBM limit the detection of GBM tumors, and current intraoperative visualization methods limit the complete tumor resection. In addition, although chemotherapy is usually used to remove residual cancer tissue after surgery, most chemotherapeutic drugs cannot effectively penetrate the BBB or enter GBM tumors. Therefore, GBM has limited treatment options and a high recurrence rate. It is necessary to improve its complete visualization during surgery and treatment [100,101,102]. In addition to MRI, near-infrared fluorescence (NIRF) imaging is also an important diagnostic method for GBM tumors. In addition to MRI, NIRF is also an important diagnostic method for GBM. Compared with MRI, NIRF can clearly distinguish the boundaries of the tumor so that the surgeon can effectively remove the tumor [103]. Combine ferumoxytol (FMX) with NIRF ligand hepthamethine (HMC) to construct HMC-FMX nano-platform. The experimental results show that HMC-FMX nanoparticles pass through the BBB and selectively aggregate in the tumor, which makes it possible to visualize the tumor tissue infiltration based on NIRF (Figure 3a) [104].

It is well known that iron oxide nanoparticle (IONP) is often used as an MRI contrast agent. In recent years, magnetic particle imaging (MPI) has attracted much attention because of its high sensitivity, and IONP can be used as a tracer of MPI. The tracer generates non-linear magnetization signals in an oscillating magnetic field for performing imaging, especially for stem cell tracking applications [105,106]. It is expected that in the near future, the development of iron-based nanomaterials can have a major breakthrough. It will also have a better application in the diagnosis and treatment of brain diseases such as AD, PD, and IS.

#### 3.2.2. Manganese-Based Nanomaterials

Manganese is an essential trace element for the human body. It is indispensable for the maintenance of normal brain function. It has a certain relationship with the development of intelligence, thinking, emotion, and behavior. Manganese is also very important for the work of the human brain and CNS and has a certain effect on AD. Interestingly, there is a superoxide dismutase in the human body, which is the most important protective enzyme in the life of the human body, and this enzyme must be catalyzed by manganese ions to function. With the development of nanotechnology, manganese-based nanomaterials are playing an increasingly important role in building an integrated platform for diagnosis and treatment.

The current Gd chelate-based MRI for stroke has relatively low sensitivity and can cause side effects, such as nephrogenic systemic fibrosis and intracranial gadolinium deposition [70,107,108,109,110]. Therefore, it is urgent to promote new and safer research imaging methods for the diagnosis and analysis of stroke. In recent years, manganese-based nanomaterials have become an integrated diagnosis and treatment nano-platform in the field of biomedicine due to their unique biological effects, such as paramagnetism, catalysis, and redox [111]. As shown in Figure 3b,c, the BSA-MnO_2_ nanoparticles prepared by Pan’s group use the one-pot method and can image the permeability of the BBB in stroke. The nanoparticle has strong imaging ability and good biocompatibility, can image the permeability of the BBB of the middle cerebral artery occlusion (MCAO) model rat in a non-invasive and timely manner, and plays a good predictive role [80]. Chen et al. used a similar protein modification strategy. They have successfully grown MnO_2_ nanocrystals on holo-transferrin (holo-Tf) in situ under milder conditions (pH = 8.4), which preserved the structure of Tf, thereby keeping it across the BBB and targeting the brain glue. Secondly, protoporphyrin (PpIX) was further loaded on holo-Tf as a sonosensitizer to obtain MnO_2_@Tf–ppIX composite nanoparticles (TMPs). Mn^2+^ can be released from TMPs in the tumor microenvironment so as to realize the enhancement of MRI signal in response to the microenvironment [40]. 

It is precisely because of the rich biological properties of Mn^2+^ and its potential as a contrast agent for MRI that manganese ion-enhanced magnetic resonance imaging (MEMRI) has applications in neuroimaging. Because toxicity is a major drawback of manganese ion contrast agents, enormous toxicity will limit its application. Manganese-based nanomaterials must be safely transported to the lesion site based on ensuring its imaging capabilities to exert their diagnostic effect [112]. An accurate diagnosis of neuroimaging will also guide the diagnosis and treatment of brain diseases.

**Figure 3 molecules-26-07340-f003:**
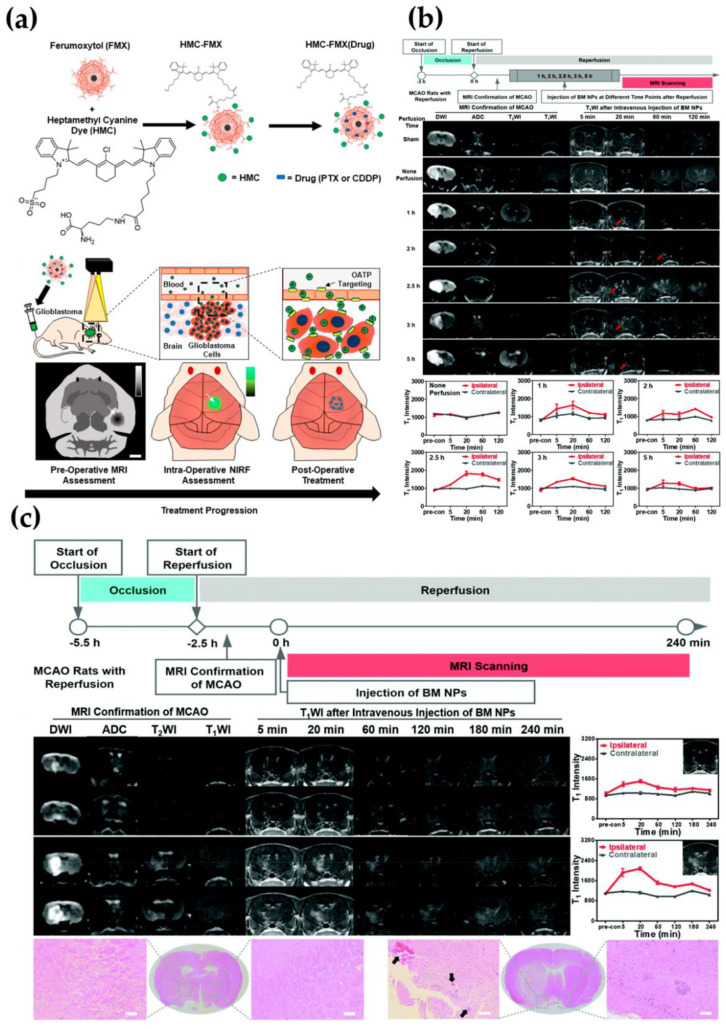
(**a**) HMC–FMX facilitates GBM tumor detection and drug delivery across the BBB, permission from [104]. Copyright (2020) American Chemical Society; (**b**) MR imaging of BBB permeability in MCAO rats without perfusion or with different perfusion time points (1, 2, 2.5, 3, and 5 h) post-injection of BM NPs, permission from [80]. Copyright (2021) Royal Society of Chemistry; (**c**) BBB permeability imaging indicated by the BM NPs in MCAO rats with or without HT, permission from [80]. Copyright (2021) Royal Society of Chemistry.

#### 3.2.3. Gold-Based Nanomaterials

In recent years, there are many types of research on gold (Au) nanomaterials. Because gold nanomaterials are chemically inert relative to other metal materials, the surface can be easily modified, and it can be synthesized into nanoclusters [113], nanorods [114], and nanostars [115]. Meanwhile, Au nanomaterials have surface plasmon resonance (SPR) characteristics, which is one of the optical properties of Au nanomaterials [116]. Therefore, it has a wide range of applications in cell imaging, ultra-sensitive detection, drug delivery, and photothermal therapy [117,118,119,120]. It is this unique optical property that makes Au nanoparticles worthy of in-depth research in imaging, such as PA [121,122,123]. In addition, Au nanoparticles can also be used as a diagnosis pattern of CT [124]. Due to the existence of BBB, the application of nanomedicine in brain diseases has been limited. Thus, Madhavan Nair et al. synthesized magnetic core/Au shell (MNP@Au) plasmonic nanoparticles with super-transient magnetism, which is helpful for MRI. In addition, the experiment also studied the migration experiment of MNP@Au nanoparticles in BBB in vitro. The experimental results show that the presence of the magnetic field increases the migration of MNP@Au nanoparticles and does not affect the integrity and permeability of the BBB [125]. In addition to magnetic targeting, using focused ultrasound to open the BBB instantaneously is also one of the better drug delivery methods [126,127]. Stanislav Y Emelianov and others synthesized silica-coated Au nanorods (Si-AuNRs) and explored the ability of the nanomaterial to be delivered to local areas of the brain under the stimulation of focused ultrasound. Furthermore, this nanomaterial has the ability of US and PA [81], which may be a powerful tool for understanding the mechanisms of neurological diseases and evaluating the effects of treatment.

Au-based nanoparticle with MRI-PA-Raman imaging function can better delineate the edge of brain tumors and has high sensitivity [128]. This method might guide the resection of brain tumors well in the future. However, Au-based nanoparticles still face some challenges, which we must solve in recent years. For example, if the concentration of Au nanoparticles is low, the effect of PA imaging will not good enough. So far, although the clinical transformation of Au-based nanoparticles has been limited, we can further upgrade the modification of the Au surface and then synthesize a suitable shape and optimize its optical effect. If we do that successfully, AuNP-based PA imaging will come true. 

#### 3.2.4. Other Metal-Based Nanomaterials

Traditional NIRF imaging mainly focuses on near-infrared imaging (NIR-I, 700–900 nm). Due to the strong absorption and high scattering of NIR-I light by biological tissues and the limited penetration depth of NIR-I light, these deficiencies have led to the need to further improve the resolution and sensitivity of NIR-I fluorescence imaging, which hinders the development of FL imaging technology in the deep tumors, such as GBM applications. Therefore, NIRF imaging has gradually expanded from NIR-I to the near-infrared second zone (NIR-II, 1000–1700 nm) with deeper penetration depth, less photon tissue attenuation, and higher temporal and spatial resolution [129,130,131]. In NIR-II imaging, rare-earth ion-doped nanoparticles are used as a new type of fluorescent nanoprobe, which has a larger Stokes shift, less biological toxicity, better photostability, and narrower emission advantages, such as spectrum. However, due to the relatively low quantum fluorescence yield of rare-earth nanoparticles, especially the long-wavelength NIR-II emission, their application in NIR-II bioimaging is limited [132,133]. According to a report, NaNdF_4_@ NaLuF_4_ nanoparticles were synthesized to enhance the fluorescence intensity [134]. This will facilitate imaging in areas of glioma. CT/MRI dual-modality imaging of the lesion area will be more conducive to the diagnosis of GBM. Wang’s research group created a polydopamine–Au nanoparticle composite platform loaded with Pifthrin-µ, which combines photothermal therapy and radiotherapy and, at the same time, enhances the proapoptotic unfolded protein response of GBM after treatment and significantly improves the therapeutic effect. The Au nanoparticles at the core of the nanoplatforms can be used for CT imaging, and the polydopamine layer on the surface can be used for MR T1-weighted imaging. The results of the study show that after the tail vein injection of the nanomaterial, CT/MRI dual-modality imaging of brain GBM can be carried out, thereby guiding radiotherapy and photothermal therapy and realizing the integration of diagnosis and treatment [135].

According to reports, cerium oxide (CeO_2_) can be used as good medicine for the treatment of brain diseases. Shi et al. used ultra-small CeO_2_ nanoparticles (3-5 nm) as the core and grafted PEG and ANG polypeptides on the surface to achieve receptor-mediated endocytosis across the BBB. The active site Ce^3+^/Ce^4+^ electron pair on the surface of the nanomaterial has a strong antioxidant capacity and can effectively scavenge free radicals. Meanwhile, the loaded edaravone (Eda) further scavenges the ROS in the IS region [136]. Doping single atoms on the surface of CeO_2_ nanoparticles is one of the effective strategies to improve their catalytic activity. According to reports, Zhang et al. synthesized a Pt/CeO_2_ nanoenzyme. Single-atom Pt caused CeO_2_ lattice expansion and preferential distribution, which greatly improved its endogenous catalytic activity and achieved the purpose of treating brain trauma [137]. In future research, we can also develop the possibility of multi-modal imaging based on CeO_2_ nanoparticles.

## 4. Summary and Outlook

In the past few decades, the development of nanomaterials and nanotechnology has gradually increased, and there has been greater progress in biomedical engineering, especially in brain diseases such as IS and AD. In the future, scientists will further accurately design the properties of nanomaterials according to the characteristics of clinical diseases so as to achieve the purpose of precise treatment. In order to clarify the structure and function of the human brain, as well as the material basis of human behavior and mental activity, clarify its mechanism at various levels, improve the efficiency of human neural activity, and improve the level of prevention, diagnosis, and treatment of neurological diseases [120,121]. In the 21st century, countries have entered the era of brain science. This will further promote research on the difficulty of nanomaterials effectively crossing the BBB so that nanomaterials achieve effective targeting across the BBB to the lesion as expected. However, we currently lack a deeper safety assessment of the pharmacokinetics, biodistribution, genetic toxicity, and reproductive toxicity of nanomaterials at the biological level. Although we are currently evaluating the safety and therapeutic effects of a series of nanomaterials at the animal level, the current sample data and research progress are difficult to advance into the clinic.

The integrated research of diagnosis and treatment of brain diseases is a new technology that integrates the diagnosis and treatment of diseases [122]. By rationally designing and synthesizing nanomaterials, integrating the current clinically independent diagnosis and treatment processes or functions into the same nanocarrier can build an integrated diagnosis and treatment platform. This not only enables a real-time and accurate diagnosis of diseases while performing disease treatments but also enables monitoring the treatment process and treatment effects, which is conducive to the development of better treatment plans [61,123]. Nanocarriers have become an ideal element of an integrated diagnosis and treatment nano-platform due to their good biocompatibility, modifiability, and excellent loading capacity. At the same time, different nanomaterials have different imaging functions, and some material systems even have multiple imaging functions.

At present, although the construction of integrated diagnosis and treatment nanomaterials faces many challenges, this intelligent diagnosis and treatment nanomaterial has great development potential. With the increasingly urgent clinical needs and the rapid development of science and technology, the development of a new diagnosis and treatment platform based on personalized precision medicine is the general trend. It is a good choice to continuously develop new imaging and treatment modes based on the optical, acoustic, and magnetic properties of nanomaterials.

## 5. Conclusions

In summary, we discussed some imaging applications of inorganic nanomaterials in different brain disease models. The use of inorganic nanomaterials with unique physicochemical characteristics can improve the imaging resolution and signal-to-noise ratio for the diagnosis of brain disease. Moreover, these non-metallic-based nanomaterials or metal-based nanomaterials can cross BBB in brain diseases by the magnetic field, focused ultrasound, receptor-mediated endocytosis, etc. We also highlighted diverse theranostic nanomaterials with multifunctional properties, which have both multi-mode imaging functions and good treatment effects. More efforts are required to develop better biocompatibility and real-time monitoring of the state of the microenvironment in the lesion area. Although the current imaging research of inorganic nanomaterials for brain diseases is still in the preliminary stage, it is also facing many challenges, such as biological safety and clinical translation. We should explore the application potential of nanomaterials on the basis of many studies and further test its underlying mechanisms. In addition to studying its targeting to different lesion sites, we should also study its ability to cross barriers. Although considering the interdisciplinary development of nanotechnology, biotechnology, and imaging omics, we believe that inorganic nanomaterials will truly move from the laboratory to clinical research to be open to the public and bring good news to patients with brain diseases. 

## Data Availability

Not applicable.

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
