# Peer review of "Inorganic Nanomaterial for Biomedical Imaging of Brain Diseases"

_molecules, 2021, doi:10.3390/molecules26237340_

Round 1

Reviewer 1 Report

The researchers have prepared an excellent review which has been prepared well. The literature survey is sufficient and supporting the main content. The inorganic nanomaterial plays an important role. This review will have good impact in the field and help researchers to learn the up-to-date status of the field. Accepted is suggested.

Author Response

Response to Reviewer 1 Comments

Point 1: The researchers have prepared an excellent review which has been prepared well. The literature survey is sufficient and supporting the main content. The inorganic nanomaterial plays an important role. This review will have good impact in the field and help researchers to learn the up-to-date status of the field. Accepted is suggested.

Response 1: We are very grateful for the reviewer's comments and recognition of our article.

Reviewer 2 Report

The main point of the article is a bit diffuse and does not focus on the nanomaterials. Instead, even biomedical imaging techniques and brain diseases are introduced on a considerable scale, which may make the reader wonder where the focus of this review is. Therefore, authors need to make appropriate modifications to their articles' content to meet the requirements for publication in Molecules.

1. The table for the characteristics of various nanomaterial-based imaging modalities in terms of spatial resolution and molecular sensitivity should be included in the manuscript.

2. The authors' title raises the reader's expectations as to whether the article will present recent developments and advances in inorganic materials for biomedical imaging. However, the first half of the article is almost redundant, presenting the established diseases and imaging techniques. It is not until page 9 that the focus of the article about nanomaterials is brought out, but the length of the presentation for these materials is also limited (only up to page 14). Therefore, it is suggested that the first half of the article could be condensed and trimmed down, as many of them are known diseases or techniques. The second half of the article needs to be expanded to include different types of inorganic materials to enrich the breadth of the article.

Author Response

Response to Reviewer 2 Comments

Point 1: The table for the characteristics of various nanomaterial-based imaging modalities in terms of spatial resolution and molecular sensitivity should be included in the manuscript.

Response 1: Thanks to the reviewer for the great comment. According to this suggestion, we added a table of the characteristics of various inorganic nanomaterials-based imaging modalities. Please see page 8, line 325-327, and line 330-331 in the revised manuscript, see also as below:

The characteristics of various inorganic nanomaterials-based imaging modalities in terms of spatial resolution and molecular sensitivity are summarized in Table 1.

Table 1 The characteristics of various inorganic nanomaterials-based imaging modalities [75]

Nanomaterial type

Imaging modality

Spatial resolution

Advantages (sensitivity, imaging

Stability, etc)

References

Nonmetallic- based

Multi-walled carbon nanotubes (f-MWNTs)

SPECT/CT

6–8 mm/0.5–0.625 mm

Fast scanning time and clear images

[76]

Hollow mesoporous silica spheres (HMSN)

MRS

-

High sensitivity

[77]

Metal-based

Fe3O4

MRI

1–2 mm

High imaging

stability and long-term

tracking 

[78, 79]

BSA-MnO2

MRI

1–2 mm

Strong imaging ability and good biocompatibility

[80]

Si-AuNRs

PA

50–200 μm

Non-invasive and non-ionizing biomedical imaging

[81]

Point 2: The authors' title raises the reader's expectations as to whether the article will present recent developments and advances in inorganic materials for biomedical imaging. However, the first half of the article is almost redundant, presenting the established diseases and imaging techniques. It is not until page 9 that the focus of the article about nanomaterials is brought out, but the length of the presentation for these materials is also limited (only up to page 14). Therefore, it is suggested that the first half of the article could be condensed and trimmed down, as many of them are known diseases or techniques. The second half of the article needs to be expanded to include different types of inorganic materials to enrich the breadth of the article.

Response 2: Thanks to the reviewer for the great comment. According to this suggestion, we cut out parts of the first and second parts of the article (line 105-109, line 116-123, line 151-156, line 173-179, line 186-191, line 251-257, line 278-279, and line 330-334 in the original manuscript). See also as below:

line 105-109:Drug therapy is the main treatment, common drugs are dopamine receptors, anticholinergics, Monoamine oxidase β inhibitors, and so on. The aim of these drugs is to prevent the neurodegeneration of dopamine. Among them, levodopa can enhance the excitability of neurons, alleviate the abnormal behavior of PD, and relieve the pain of patients to some extent [14].”

line 116-123:” Taking N-isobutyryl-L-cysteine (L-NIBC) modified AuNCs as an example, they found that AuNCs can effectively inhibit α-Syn protein fibrosis in vitro experiments; in cell experiments, it has shown that it can effectively inhibit MPP+ induced PD model cells. More importantly, the experimental results of the mouse PD model showed that AuNCs greatly improved the behavioral disorders of the diseased mice. Immunomics and western blot analysis showed that AuNCs can significantly reduce the death of dopa-minergic neurons in the substantia nigra and striatum of diseased mice.

line 151-156:” It has the four characteristics of high morbidity, high mortality, high disability, and high recurrence rate. IS is caused by atherosclerosis, thrombosis, and foreign bodies in the arteries that supply blood to the brain, which leads to stenosis or even complete blockage of the vascular lumen, which affects blood circulation and leads to the insuf-ficient blood supply and oxygen supply in local brain tissue [23].

line 173-179:Eliminating excess ROS and preventing its destruction and damage to brain cells has important scientific significance in the fields of clinical medicine basic research and applied research. On the other hand, the main obstacle to the treatment of stroke with drugs in the body is the difficulty in crossing the BBB. BBB is a physiological structure that controls the transport and delivery of molecules from blood vessels to the brain parenchyma and maintains a delicate balance in the brain [30, 31].

line 186-191:” Studies have shown that the nasal administration of nanoliposomes carrying basic fi-broblast growth factors can significantly increase the content of basic fibroblast growth factors in mice with IS, thereby having a certain nerve repair effect [32]. Ling et al. re-ported a new strategy to efficiently improve embryo-derived natural mesenchymal stem cells (MSCs) using magnetic iron oxide nanochains.

line 251-257:” MRI applies radio frequency pulses of a certain frequency to the human body in a static magnetic field, so that the hydrogen protons in the human body are excited to cause magnetic resonance. After the pulse is stopped, the proton generates a magnetic reso-nance signal during the relaxation process. Magnetic resonance signals are generated through processing procedures such as receiving, spatial coding, and image recon-struction of magnetic resonance signals. MRI has been used in imaging the diagnosis of various systems throughout the body.

line 278-279:” The high-energy radiation source used in the CT imaging process can cause irreversible damage to human tissues [50].

line 330-334:” US is commonly used to determine the location, size, and shape of organs, determine the scope and physical properties of the lesions, provide some anatomical diagrams of glandular tissues, and distinguish between normal and abnormal fetuses. In ophthal-mology, obstetrics and gynecology, cardiovascular system, Digestive system, and uri-nary system are widely used [69, 70].

We also added some elaboration in part three. Please see page 10, line 396-404 in the revised manuscript, see also as below:

“Especially, Fe3O4 and maghemite (γ-Fe2O3)-based crystalline particles can be prepared into nanomaterials (3-100 nm) with different characteristics by controlling raw materials, reaction time, and preparation methods. Because both iron oxides contain atomic holes and surface defects, they are easy to be modified for surface functionalization [94-96]. Interestingly, Fe3O4 prepared by conventional methods is magnetic, has a cubic inverse spinel structure, and often has tetrahedral and octahedral positions. After reasonable modification, it can be better dispersed to form a magnetic fluid, which can be imaged under the condition of a magnetic field [97].”

Please see page 11, line 433-440 in the revised manuscript, see also as below:

“It is well known that iron oxide nanoparticle (IONP) is often used as an MRI con-trast agent. In recent years, magnetic particle imaging (MPI) has attracted much attention because of its high sensitivity, and IONP can be used as a tracer of MPI. The tracer generates nonlinear magnetization signals in an oscillating magnetic field for per-forming imaging, especially for stem cell tracking applications [105, 106]. It is expected that in the near future, the development of iron-based nanomaterials can have a major breakthrough. It will also have a better application in the diagnosis and treatment of brain diseases such as AD, PD, and IS.”

Please see page 13, line 531-537 in the revised manuscript, see also as below:

“It is precise because of the rich biological properties of Mn2+ and its potential as a contrast agent for MRI that manganese ion-enhanced magnetic resonance imaging (MEMRI) has applications in neuroimaging. Because toxicity is a major drawback of manganese ion contrast agents, enormous toxicity will limit its application. Manganese-based nanomaterials must be safely transported to the lesion site based on ensuring its imaging capabilities to exert their diagnostic effect [112]. Accurate diagnosis of neuroimaging will also guide the diagnosis and treatment of brain diseases.”

Please see page 14, line 546-585 in the revised manuscript, see also as below:

3.2.3 Gold-based nanomaterials

In recent years, there are many types of research towards gold (Au) nanomaterials. Because gold nanomaterials are chemically inert relative to other metal materials, the surface is easy to be modified, and it can be synthesized into nanoclusters [113], nanorods [114], and nanostars [115]. Meanwhile, Au nanomaterials have the characteristic of surface plasmon resonance (SPR) characteristics, which is one of the optical properties of Au nanomaterials [116]. Therefore, it has a wide range of applications in cell imaging, ultrasensitive detection, drug delivery, and photothermal therapy [117-120]. It is this unique optical property that makes Au nanoparticles worthy of in-depth re-search in imaging, such as PA [121-123]. In addition, Au nanoparticles can also be used as diagnosis pattern of CT [124]. Due to the existence of BBB, the application of nano-medicine in brain diseases has been limited. Thus, Madhavan Nair et al. synthesized magnetic core/Au shell (MNP@Au) plasmonic nanoparticles with super-transient magnetism, which is helpful for MRI. In addition, the experiment also studied the migration experiment of MNP@Au nanoparticles in BBB in vitro. The experimental results show that the presence of the magnetic field increases the migration of MNP@Au nanoparticles, and does not affect the integrity and permeability of the BBB [125]. In addition to magnetic targeting, using focused ultrasound to open the BBB instantaneously is also one of the better drug delivery methods [126, 127]. Stanislav Y Emelianov and others synthesized silica-coated Au nanorods (Si-AuNRs) and explored the ability of the nanomaterial to be delivered to local areas of the brain under the stimulation of focused ultrasound. Furthermore, this nanomaterial have the ability of US and PA [81], which may be a powerful tool for understanding the mechanisms of neurological diseases and evaluating the effects of treatment.

Au-based nanoparticle with MRI-PA-Raman imaging function can better delineate the edge of brain tumors and has high sensitivity [128]. This method maybe well guides the resection of brain tumors in the future. But Au-based nanoparticles still face some challenges, which we must solve in recent years. For example, if the concentration of Au nanoparticles is low, the effect of PA imaging will not good enough. So far, although the clinical transformation of Au-based nanoparticles has been limited, we can further upgrade the modification of the Au surface, and then synthesize a suitable shape and optimize its optical effect. If we do that successfully, Au NP-based PA imaging will come true.”

Reviewer 3 Report

1.". The use of the special physical and chemical properties of inorganic nanomaterials can enhance the contrast between the patient's lesion and surrounding tissues, which can promote the development of biomedicine to a certain extent. " - ONLY THE WORD CAN? NO CERTAIN CONCLUSION?

2."Because the application of bioimaging technology will improve the clarity and resolution of traditional imaging, clinicians will obtain richer and more accurate imaging information, which is conducive to disease diagnosis and prescriptions."  ONLY THE WORD WILL? NO CERTAIN CONCLUSION?

3."In the future, it is necessary to develop more functional multi-modal contrast agents for real-time monitoring of the state of the cell microenvironment in the lesion area" - FUTURE? 

4."However, considering the interdisciplinary development of nanotechnology, biotechnology, and imaging omics, we believe that in the near future, inorganic nanomaterials will truly move from the laboratory to clinical research to be open to the public and bring good news to patients". - AGAIN FUTURE.

5." Although the current imaging research of inorganic nanomaterials for brain diseases is still in the preliminary stage, it is also facing many challenges such as biological safety and clinical translation". 

- FROM ALL THESE CHALLANGES PLEASE WRITE A CONCLUSION.

Author Response

Response to Reviewer 3 Comments

1.". The use of the special physical and chemical properties of inorganic nanomaterials can enhance the contrast between the patient's lesion and surrounding tissues, which can promote the development of biomedicine to a certain extent. " - ONLY THE WORD CAN? NO CERTAIN CONCLUSION?

Response 1: Thanks to the reviewer for the great comment. According to this suggestion, we rewrite more specifically as below:

The use of inorganic nanomaterials with unique physicochemical characteristics can improve the imaging resolution and signal-to-noise ratio for the diagnosis of brain disease. Moreover, these nonmetallic-based nanomaterials or metal-based nanomaterials can cross BBB in brain diseases by magnetic field, focused ultrasound, receptor-mediated endocytosis, etc.

2." Because the application of bioimaging technology will improve the clarity and resolution of traditional imaging, clinicians will obtain richer and more accurate imaging information, which is conducive to disease diagnosis and prescriptions."  ONLY THE WORD WILL? NO CERTAIN CONCLUSION?

Response 2: Thank you very much for your wonderful comment. According to your suggestion, we rewrite the sentence. Please see as below:

We also highlighted diverse theranostic nanomaterials with multifunctional properties, which have both multi-mode imaging functions and good treatment effects.

3." In the future, it is necessary to develop more functional multi-modal contrast agents for real-time monitoring of the state of the cell microenvironment in the lesion area" - FUTURE? 

Response 3: We appreciate your suggestion, which could make our conclusion more reasonable. Please see also as below:

More efforts are require to develop better biocompatibility and real-time monitoring of the state of the microenvironment in the lesion area.

4." However, considering the interdisciplinary development of nanotechnology, biotechnology, and imaging omics, we believe that in the near future, inorganic nanomaterials will truly move from the laboratory to clinical research to be open to the public and bring good news to patients". - AGAIN FUTURE.

Response 4: Thank you very much for your wonderful comment. According to your suggestion, we change the logic of the sentence. Please see as below:

Although the current imaging research of inorganic nanomaterials for brain diseases is still in the preliminary stage, it is also facing many challenges such as biological safety and clinical translation.

5." Although the current imaging research of inorganic nanomaterials for brain diseases is still in the preliminary stage, it is also facing many challenges such as biological safety and clinical translation". 

- FROM ALL THESE CHALLANGES PLEASE WRITE A CONCLUSION.

Response 5: Thanks to the reviewer for the great comment. According to this suggestion, we added a conclusion after the sentence. Please see as below:

We should explore the application potential of nanomaterials on the basis of many studies, and further test its underlying mechanisms. In addition to studying its targeting to different lesion sites, we should also study its ability to cross barriers.

We believe that by addressing your suggestions and recommendations, the quality of our paper has been significantly improved. Again, we greatly appreciate your comments and evaluation of this paper.

Round 2

Reviewer 2 Report

The authors have addressed the comments properly. I would suggest acceptance of the manuscript. 

Reviewer 3 Report

The final decision is up to the Editor!